# Disruptions of *rpiAB* Genes Encoding Ribose-5-Phosphate Isomerases in *E. coli* Increases Sensitivity of Bacteria to Antibiotics

**DOI:** 10.3390/cells13221915

**Published:** 2024-11-19

**Authors:** Tatyana A. Seregina, Rustem S. Shakulov, Svetlana A. Sklyarova, Alexander S. Mironov

**Affiliations:** Engelhardt Institute of Molecular Biology, Russian Academy of Science, 119991 Moscow, Russia; rshakulov@yandex.ru (R.S.S.); sklyarovasveta@yahoo.com (S.A.S.); alexmir_98@yahoo.com (A.S.M.)

**Keywords:** pentose phosphate pathway, ribose-5-phosphate isomerases RpiA and RpiB, ribose-5-phosphate, antibiotics, *Escherichia coli*

## Abstract

In *Escherichia coli* cells, the main enzymes involved in pentose interconversion are ribose-5-phosphate isomerases RpiA and RpiB and ribulose-5-phosphate epimerase Rpe. The inactivation of *rpiAB* limits ribose-5-phosphate (R5P) synthesis via the oxidative branch of the pentose phosphate pathway (PPP) and unexpectedly results in antibiotic supersensitivity. This type of metabolism is accompanied by significant changes in the level of reducing equivalents of NADPH and glutathione, as well as a sharp drop in the ATP pool. However, this redox and energy imbalance does not lead to the activation of the *soxRS* oxidative stress defense system but the increased sensitivity to oxidants paraquat and H_2_O_2_. The deletion of *rpiAB* leads to a significant increase in the activity of transketalase (Tkt), a key enzyme of the nonoxidative branch of the PPP and increased sensitivity to ribose added in the growth medium. The phenotype of supersensitivity of *rpiAB* to antibiotics and ribose can be suppressed by activating the utilization of sedoheptulose-7-phosphate, which originates from R5P, to LPS synthesis or limitation of nucleoside catabolism by the inactivation of the DeoB enzyme, responsible for conversion of ribose-1-phospate to R5P. Our results indicate that the induction of unidirectional synthesis of R5P is the cause of supersensitivity to antibiotics in *rpiAB* mutant.

## 1. Introduction

In the last decade, many studies have appeared devoted to the investigation of the nonspecific effect of antibiotics on the metabolic status of the bacterial cell [1]. It has been shown that the sensitivity of cells to the action of antibiotics is largely determined by changes in biosynthetic processes, respiration, energy exchange, and the regulation of redox balance [2,3,4,5,6,7,8]. Discovering the key points in a branched metabolic network, the influence of which allows modules to increase sensitivity to antibacterial therapy in resistant pathogens, is a promising direction of research for the creation of highly effective drugs of a new generation.

Pentose phosphates in cells are formed as a result of reactions of the pentose phosphate pathway (PPP) [9]. In the oxidative branch of the canonical PPP, the irreversible oxidation of glucose-6-phosphate occurs, followed by the decarboxylation of phosphogluconate to ribulose-5-phosphate, accompanied by the reduction in NADP^+^ equivalents (Figure 1a) [10]. The non-oxidative branch of the PPP is reversible and serves to return excess pentoses to glycolysis (Figure 1a). The imbalance in this metabolic node leads to the development of a redox imbalance, sensitivity to oxidants, and also to a change in the interconversion of pentose phosphates (PPs) [10,11,12,13]. PPs synthesis is performed in the oxidative branch of PPP and is associated with the generation of NADPH equivalents, while the non-oxidative branch serves to return excess pentoses to glycolysis. Thus, the cell can generate the required amount of reducing agents without producing an excess of PPs. The key role in the mutual conversion of PPs is played by ribose-5-phosphate isomerases (*rpiA* and *rpiB*) and ribulose-5-phosphate epimerase (*rpe*) (Figure 1a).

The inactivation of isomerase genes leads to a radical reorganization of metabolic flows in the PPP (Figure 1b) [13]. It should be noted that the RpiA isomerase in *E. coli* cells works predominantly towards the synthesis of R5P, while RpiB carries out the opposite reaction [14]. In addition, it has been shown that the preferred substrate for RpiB is D-allose [15]. Thus, RpiA plays a major role in the synthesis of R5P in *E. coli* cells. A partner of isomerases in the mutual interconversion of pentoses is the epimerase Rpe, which controls the reversible conversion of ribulose-5-phosphate (Ru5P) to xylose-5-phosphate (Xu5P) (Figure 1a).

The cofactor of Rpe is Fe^2+^, which makes this enzyme vulnerable under conditions of oxidative stress [16]. The inactivation of epimerase deprives cells of the ability to use PPs as the only sources of carbon and energy [17].

Despite the fact that the synthesis of PPs is one of the most important anabolic processes necessary for cell growth and division, the effect of inactivation of ribose-5-phosphate isomerases and epimerase genes on the redox status, as well as the central metabolism of the bacterial cell, remains poorly understood. This work is devoted to studying the effect of deletions of the *rpiAB* and *rpe* genes, encoding isomerases and epimerase in *E. coli* cells on the sensitivity of bacteria to antibiotics and on changes in redox balance. The process of pentose phosphate synthesis can be considered as a target for the development of new types of antimicrobial agents.

## 2. Materials and Methods

### 2.1. Bacterial Strains

The bacterial strains of *E. coli* used in the work and their genotypes are presented in Table 1.

Deletion mutants were obtained by growing phage P1 on strains from the Keio collection [20] containing the insertions *rpiA*::kan, *rpiB*::kan, *rpe*::kan, *deoB*::kan, and *zwf*::kan and their subsequent transduction into the genome of strain *E. coli* MG1655. The kanamycin cassette was removed from the resulting strains using the helper plasmid pCP20 [18] with the formation of the corresponding deletions *∆rpiA, ∆rpiB, ∆rpe, ∆deoB*, and ∆*zwf*. The presence of deletions was confirmed by PCR experiments. The production of a strain containing the *gmhA* gene under the control of the constitutive P_tet_ promoter was carried out by analogy with work [7].

To quantitatively assess the level of activation of the *soxRS* system genes, we used the hybrid plasmid pSoxS’::*lux*, in which the promoter-operator region in front of the *soxS* gene is transcriptionally fused with the *luxCDABE* gene cassette of *P. luminescens* [19].

### 2.2. Media and Growing Conditions

The Luria–Bertani (LB) medium was used as a complete nutrient medium for growing bacteria. In a liquid medium, bacteria were cultivated on a shaker (200 rpm) at 37 °C. If necessary, the following were added to the medium: nalidixic acid (25 µg/mL), moxifloxacin (3 µg/mL), gentamicin (3 µg/mL), erythromycin (150 µg/mL), rifampicin (25 µg/mL), ampicillin (100 µg/mL), kanamycin (20 µg/mL), hydrogen peroxide (2.5 mM), paraquat (250 µM), D-ribose (0.1%), and D-xylose (0.1%). All reagents used in the work were produced by Sigma-Aldrich, St. Louis, MO, USA, unless otherwise stated.

### 2.3. Generation of Growth Curves

Growth curves were plotted using an automatic Bioscreen C device (Oy Growth Curves Ab Ltd., Turku, Finland). Overnight cultures of bacteria grown at 37 °C in LB medium were diluted 100 times, placed in the wells of the Bioscreen device platform, and grown at 37 °C with maximum shaking. OD_600nm_ values were recorded automatically at certain time intervals. Each experiment was carried out in triplicate and the average values OD_600nm_ were used to construct bacterial growth curves.

### 2.4. Determination of Bacterial Sensitivity to Antibiotics and Oxidizing Agents

Overnight bacterial cultures were diluted 100 times and grown with aeration at 37 °C to optical density OD_600nm_ ≈ 0.5, treated with the indicated concentrations of antibiotics and oxidizing agents and continued to grow for 30 min, then dilutions were made and, plated on Luria–Bertani broth agar plates, which were placed in a thermostat at 37 °C for 24 h. Survival was determined by counting colonies in three independent experiments to determine average values. In addition, colony formation ability was assessed by microdilution method. Overnight bacterial cultures were diluted 100 times and grown on a thermostatic shaker at 37 °C to optical density OD_600nm_ ≈ 0.5. All suspensions were equalized by optical density. A series of tenfold dilutions were prepared from the resulting cultures in a 96-well plate in a volume of 100 µL. The resulting dilutions were sown on plates with rich medium containing various concentrations of the antibiotics under study. The dishes were incubated overnight in a thermostat at 37 °C. The result was photographed using the GelCamera M-26XV Analytical System (UVP, Upland, CA, USA).

### 2.5. Measuring NADPH Levels

NADPH levels were measured using a fluorimetric NADP/NADPH Assay Kit (Abcam Laboratories, Chicago, IL, USA). Cells were grown to OD_600nm_ ≈ 0.5 in a thermostatic shaker at a temperature of 37 °C. The preparation of cell extracts, as well as all subsequent manipulations, were carried out according to the manufacturer’s instructions included in the kit. Fluorescence detection of the samples was carried out in a Tecan Spark plate reader (Tecan Grup LTD, Männedorf, Switzerland) at Ex/Em = 540/590 nm. The results obtained were related to the optical density of the culture OD_600nm_ and expressed as a percentage. The NADPH level in the wild-type strain was taken as 100%.

### 2.6. Measuring Intracellular Glutathione Levels

Quantitative determination of intracellular glutathione levels was carried out using a modified Titz method [21]. An amount of 100 μL of the overnight culture was transferred to 10 mL of fresh LB medium and grown to OD_600nm_ ≈ 0.5. Cells from 5 mL of suspension were pelleted by centrifugation and suspended in KPE lysis buffer with the addition of 0.1% Triton X100. Cells were homogenized with a pestle and centrifuged at 11,000 rpm for 5 min at 4 °C. The super was transferred into clean test tubes. A reaction mixture consisting of 20 μL of cell extract and equal volumes (60 μL) of DTNB, glutathione reductase and NADPH was incubated for 3 min at room temperature, after which adsorption was measured at 412 nm. GSH (Sigma-Aldrich, St. Louis, MO, USA) was used to construct a calibration curve. The obtained values OD_600nm_ were referred to the optical density of the cultures. To determine the oxidized form of glutathione (GSSG), the resulting cell extracts were treated with 2-vinylpyridine as described in [21]. Oxidized glutathione (Sigma-Aldrich, St. Louis, MO, USA) was used to construct a calibration curve.

### 2.7. ATP Level Measurement

The intracellular ATP level was determined by the luminescent method using the Adenosine 5′-triphosphate (ATP) Bioluminescent Assay Kit (Sigma-Aldrich, St. Louis, MO, USA) on a Tecan Spark plate reader (Tecan Grup LTD, Männedorf, Switzerland). An mount of 5 mL of cell suspension grown to OD_600nm_ ≈ 0.5 was rapidly washed twice with saline and suspended in 2 mL of saline. The reaction mixture was prepared and luminescence measurements were carried out as described in the manufacturer’s instructions. The obtained values OD_600nm_ were referred to the optical density of the culture and expressed as a percentage. The ATP level in the wild-type strain was taken as 100%.

### 2.8. Determination of soxS Promoter Activity

Overnight cultures of bacteria containing the pSoxS’::lux plasmid were diluted 100 times in fresh LB medium and grown with aeration at 37 °C until the early exponential stage. Samples of 200 μL were transferred into a 96-well plate. The detection of the luminescence signal was carried out on a Tecan Spark tablet reader for 3 h at a wavelength of 495 nm [19].

### 2.9. Determination of Transketolase Activity

Transketolase activity was measured using the Transketolase Activity Assay Kit (Fluorometric) (Abcam Laboratories, Chicago, IL, USA). Cells were concentrated in 100 µL of TKT Assay Buffer to a density of 1 × 10^6^ per mL. Further manipulations were performed in accordance with the manufacturer’s recommendations. The obtained values of Tkt activity were related to the concentration of total protein in the studied cell extracts.

### 2.10. Statistical Analysis

The data are shown as mean ± standard deviation measures from triplicate values obtained from 3 to 4 independent experiments (Appendix A, Table A1, Table A2 and Table A3). The statistical difference between experimental groups was analyzed by one-way ANOVA with Tukey correction for multiple comparisons. Probability values (*p*) less than 0.05 were considered significant. Statistical analysis was performed using the GraphPad Prism 9.1.2 software (released on 2021) (GraphPad Software Inc., San Diego, CA, USA).

## 3. Results

### 3.1. Inactivation of RpiAB Isomerase Genes Potentiates the Effect of Antibiotics

The Inactivation of ribose-5-phosphate isomerases and epimerase genes in bacteria limits the usage of PPs as carbon sources [14,17]. However, such modifications of PPP should entail more dramatic changes in central metabolism. In the present study, deletion mutants of *E. coli* for the *rpiAB* isomerase gene were constructed to determine the contribution of this enzyme to the processes of PP reversal.

The simultaneous inactivation of the *rpiA* and *rpiB* isomerase genes leads to a sharp increase in the sensitivity of *E. coli* cells to various antibiotics, while cells with a deletion of the *rpe* gene differ slightly from wild-type cells (Figure 2a,b). The cultivation of a serial of tenfold cell dilutions on a solid medium with the addition of the indicated concentrations of antibiotics revealed a significant decrease in colony formation in the *rpiAB* mutant (Figure 2a). Time-killing experiments in the presence of high concentrations of antibiotics also showed a decrease in tolerance of *rpiAB* mutants compared to the parent strain and the *rpe* mutant (Figure 2b).

It should be noted that turning off the stage of conversion of Ru5P to R5P radically changes the process of R5P synthesis, leading to the reversion of the non-oxidative branch of PPP (Figure 1b). The *rpe* mutation does not lead to the separation of the oxidative and non-oxidative branches of the PPP, since the additional formation of Xu5P from F6P and GAP with the participation of transketolase Tkt is possible (Figure 1b). Thus, we propose that the process of ribose synthesis plays a key role in the development of sensitivity to antibiotics in *rpiAB* mutant.

### 3.2. Characteristics of RpiAB and Rpe Mutants Associated with PPP Function

NADPH and glutathione levels are indicators of redox homeostasis and regulate the activity of the oxidative branch of the PPP, while the ATP pool determines the anabolic potential of the cell [22]. The deletion of the *rpiAB* genes leads to a slight decrease in the level of reduced NADPH equivalents, while the ∆*rpe* mutant does not differ from the parent strain (Figure 3a).

One of the nonspecific actions of antibiotics is the induction of oxidative stress [2]. NADPH equivalents are involved in the restoration of another important protector against oxidative stress—glutathione. Accordingly, a significant decrease in the total glutathione pool is observed in the ∆*rpiAB* mutant. In addition, the ratio between the reduced and oxidized forms of glutathione is shifted towards the latter (Figure 3b). Thus, in the isomerase gene mutant *rpiAB*, we observe a significant deficiency of reducing agents, which indicates the occurrence of a redox imbalance. Furthermore, we observed a dramatic decrease in the ATP pool in ∆*rpiAB* mutants (Figure 3c), which correlates with a low level of total glutathione.

The activity of the oxidative branch of PPP is regulated by the pSoxS promoter of the *zwf* gene, encoding glucose-6-phosphate dehydrogenase, which carries out the first step in the conversion of glucose-6-phosphate into 6-phosphogluconolactone [12]. Thus, PPP is one of the key components of the *soxRS* system of response to oxidative stress. The activity of the *soxS* promoter is a marker of oxidative stress [12].

To determine the level of activity of the *soxRS* system, mutants ∆*rpiAB* and ∆*rpe* were transformed with a plasmid, which is a lux-biosensor based on the *lux*-operon of luminous bacteria under the control of the *soxS* promoter. Surprisingly, we did not found activation of *soxS* promoters in *E. coli* cells, containing deletions of *rpiAB* and *rpe* (Figure 4a). A strain with a deletion of the *zwf* gene, which provokes oxidative stress, was used as a positive control [12].

The testing of *rpiAB* and *rpe* mutants for sensitivity to oxidizing agents such as hydrogen peroxide and paraquat did not reveal significant differences from the wild-type strain (Figure 4b). Apparently, the redox imbalance observed in the *rpiAB* mutant is different from the canonical oxidative stress response.

### 3.3. Unidirectional Ribose-5-Phasphate Synthesis Is Crucial Factor for Cell Growth and Antibiotic Lethality

Based on the assumption that *rpiAB* deletions lead to the reversion of the non-oxidative branch of PPP for the synthesis of R5P and disrupt the normal interconversion of PPs in the cell, an experiment was carried out to study the effect of PPs on the growth of the *rpiAB* mutant culture. It should be noted that the absence of ribose-5-phosphate isomerase activity significantly slows down cell growth compared to the parent strain and the *rpe* mutant (Figure 5a).

The inactivation of ribose-5-phosphate isomerase genes makes the synthesis of R5P impossible via the oxidative branch (Figure 1b). Thus, the cell faces the necessity of reversing the non-oxidative branch of the PPP, where transketolase Tkt plays a key role. More than a threefold increase in the activity of this enzyme was found in the *rpiAB* mutant (Figure 5b), whereas the deletion of *rpe* results in a strong decrease in transketolase activity (Figure 5b).

Remarkably, the addition of D-ribose to the growth medium significantly inhibited the growth of the *rpiAB* mutant, whereas D-xylose had no such effect (Figure 5c). The ∆*rpe* mutant in this experiment and did not show significant changes in growth (Figure 5c).

Summarizing the above results, we assumed that the R5P pool plays a key role in the sensitivity of the *rpiAB* mutant to antibiotics. The formation of the ribose pool involves two opposing processes: de novo synthesis via PPP and the catabolism of purine and pyrimidine ribonucleosides. Thus, limiting the catabolism of nucleosides as a source of R5P or increasing the consumption of precursor metabolites such as sedoheptulose 7-phosphate from PPP should suppress of the *rpiAB* sensitivity phenotype (Figure 1b).

Indeed, inactivation of phosphopentomutase DeoB, responsible for the conversion of R1P to R5P [23] or the increased synthesis of the cell wall component ADP-heptose via P_tet_-*gmhA* [24] leads to the suppression of the sensitivity of the ∆*rpiAB* mutant to antibiotics (Figure 6a,b). In addition, the deletion of *deoB* and the overexpression of *gmhA* restore the tolerance of *rpiAB* cells to excess D-ribose in the growth medium (Figure 6c).

## 4. Discussion

Our data indicate that ribose-5-phosphate isomerase activity is a switch between the katabolic synthesis of R5P from glucose with the possibility of returning its excess to glycolysis and a one-way anabolic process, which apparently does not allow for the clear regulation of the R5P pool in the cell (Figure 1). It has been previously shown that the revers of the non-oxidative branch of the PPP occurs during the inactivation of the enzymes of the oxidative branch of the PPP, the *zwf* and *gnd*, which leads to redox imbalance and decreased stress resistance [25]. This work demonstrates for the first time the effect of deletions of ribose-5-phosphate isomerase genes, enzymes of the non-oxidative branch РРР, on sensitivity to a wide range of antimicrobial drugs. In the *rpiAB* mutant, a new type of metabolism arises, characterized by the reversion of the non-oxidative branch of the PPP and, as a consequence, the uncoupling of the generation of reducing equivalents of NADPH from the synthesis of R5P. The regulation of the level of PPs synthesis and, notably, ribose, is carried out by the level of NADP^+^, and can be enhanced under oxidative stress conditions by an increasing level of oxidized glutathione (GSSG) [22]. A redox imbalance and restriction of anabolic capacity in the *rpiAB* mutant is supported by findings of reduced NADPH equivalents, glutathione, and ATP pool, indicating low metabolic flux through the oxidative branch.

However, such redox changes do not lead to the classical response, i.e., the activation of the *soxRS* defense system, as well as increased sensitivity to the oxidants paraquat and hydrogen peroxide [26]. Comparing the phenotypes of the ∆*rpiAB* and ∆*rpe* mutants revealed significant differences not only in the sensitivity to antibiotics and redox parameters (Figure 2 and Figure 3), but also in the activity of transketalase (Tkt) which is the key enzyme of the non-oxidative branch of the PPP (Figure 5c). A more than threefold increase in Tkt activity in Δ*rpiAB*, in addition, is accompanied by intolerance to excess D-ribose in the growth medium, which we do not observe in the *rpe* mutant (Figure 5).

Based on the assumption that the *rpiAB* phenotype is due to the inability to regulate the R5P pool due to the unidirectionality of its synthesis, we showed that the sensitivity of the *rpiAB* mutant to antibiotics can be suppressed by increasing the outflow of intermediates of the non-oxidative branch (P_tet_-*gmhA*) and limiting the catabolism of nucleosides (*ΔdeoB*), as an additional source of R5P.

## 5. Conclusions

In summary, our study demonstrates that the functional state of ribose-5-phosphate isomerase determines the direction of ribose synthesis in *E. coli* cells and the exceptional role of this process in potentiating the action of various antibacterial agents. The mechanism of this sensitivity, as well as the possible direct toxic effect of excess R5P on cells, requires further research. The synthesis of pentose phosphates, as one of the main metabolic processes, has great potential for the creation of new antibacterial drugs or adjuvants to previous generation antibiotics, as well as various therapeutic agents for the treatment of oncological diseases.

## Figures and Tables

**Figure 1 cells-13-01915-f001:**
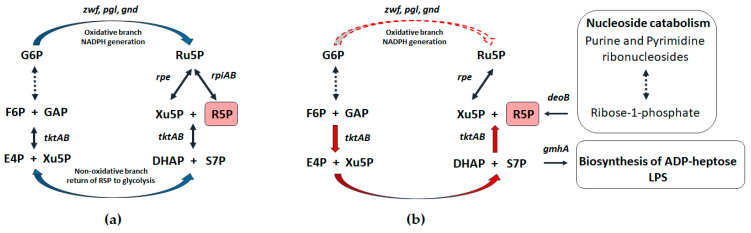
(**a**) Scheme of canonical pentose phosphate pathway (PPP). Isomerases RpiA and RpiB convert ribulose-5-phosphate (Ru5P) to R5P. Rpe epimerase catalyzes the reaction that converts Ru5P to xylose 5-phosphate (Xu5P). As a result of the synthesis of R5P in the oxidative branch of PPP, the reduction in NADP^+^ equivalents occurs. Excess PPs can be returned to glycolysis through the non-oxidative branch of PPP. (**b**) Inactivation of *rpiA* and *rpiB* genes does not lead to cell death since the synthesis of R5P from the glycolysis products fructose-6-phosphate (F6P) and glyceraldehyde-3-phosphate (GAP) is possible by reversing the non-oxidative branch of PPP. Abbreviations used: G6P—glucose-6-phosphate; 6PG—6-phosphogluconate; Ru5P—ribulose-5-phosphate; R5P—ribose-5-phosphate; Xu5P—xylose-5-phosphate; S7P—sedoheptulose-7-phosphate; DHAP—dihydroxyacetone phosphate; E4P—erythrose-4-phosphate; F6P—fructose-6-phosphate; GAP—glyceraldehyde-3-phosphate.

**Figure 2 cells-13-01915-f002:**
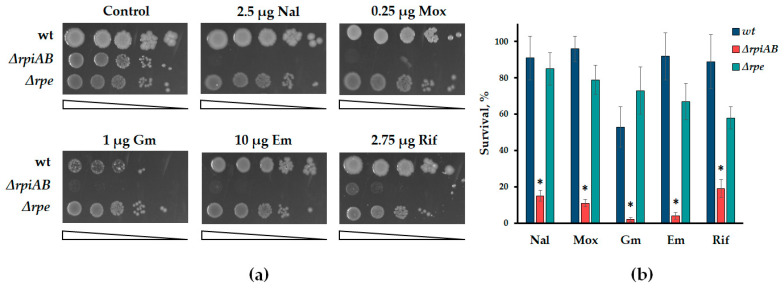
Sensitivity of *E. coli* strains with inactivated *rpiAB* and *rpe* genes to various antibiotics. (**a**) Representative efficiencies of colony formation of WT (MG1655) and mutant *E. coli* cells in the presence of antibiotics: quinolones (nalidixic acid (Nal) and moxifloxacin (Mox)), aminoglycosides (gentamicin (Gm), macrolides (erythromycin (Em)), inhibitors RNA polymerase (rifampicin (Rif)). Cells were spotted on LB agar plates in serial 10-fold dilutions and incubated at 37 °C for 24 h; (**b**) cell survival was determined by counting cfu and is shown as the mean ± SD from three independent experiments. * *p* < 0.05, compared to the wild-type cells. Overnight cultures of indicated *E. coli* strains were diluted with fresh LB 1:100 and grown to OD_600nm_ ≈ 0.5. Antibiotics were added for 30 min at concentration 25 µg/mL Nal, 3 µg/mL Mox, 3 µg/mL Gm, 150 µg/mL Em and 25 µg/mL Rif.

**Figure 3 cells-13-01915-f003:**
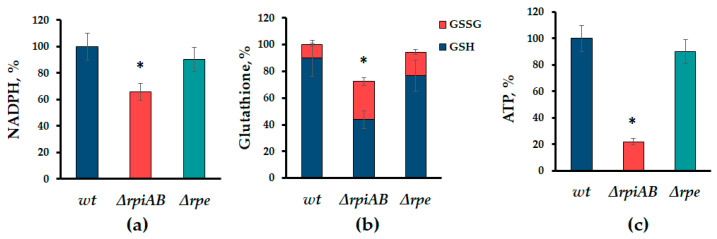
Levels of reduced equivalents of (**a**) NADPH, (**b**) glutathione, and (**c**) ATP in the ∆*rpiAB* and ∆*rpe* mutants. Mean values ± SD from at least three independent experiments are shown. * *p* < 0.05, compared to the wild-type cells.

**Figure 4 cells-13-01915-f004:**
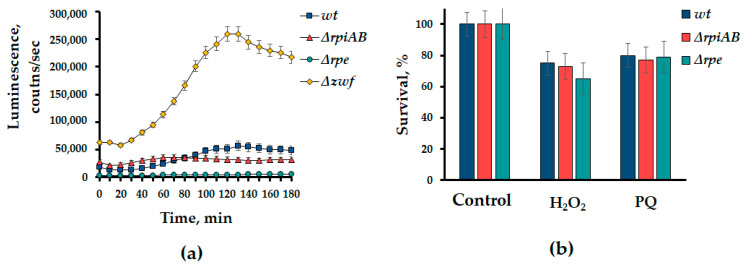
*∆rpiAB* and *∆rpe* mutants exhibit resistance to oxidative stress. (**a**) Luminescence intensity of the lux-biosensor containing the *soxS* promoter in the ∆*rpiAB* and ∆*rpe* strains. The *∆zwf* strain was used as a positive control; (**b**) cell survival of ∆*rpiAB* and ∆*rpe* strains in the presence of hydrogen peroxide (2.5 mM) and paraquat (PQ) (250 µM). Cell suspensions were incubated in the presence of oxidizing agents for 30 min at 37 °C and plated on plates with solid medium to count colonies. Mean values ± SD from at least three independent experiments are shown.

**Figure 5 cells-13-01915-f005:**
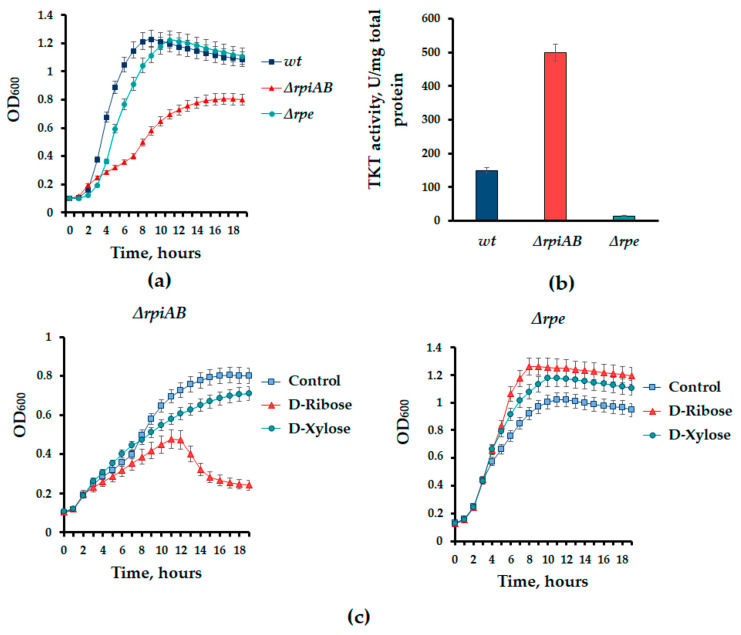
(**a**) Growth curves of the parent strain MG1655 (wt) and mutants *rpiAB* and *rpe*. (**b**) Representative curves demonstrating the effect of exogenous pentoses on the growth of *rpiAB* and *rpe* mutants. Cells were grown in complete LB medium supplemented with 0.1% ribose or xylose. (**c**) Level of transketolase activity in cell extracts of the parental strain and mutants. Mean values ± SD from at least three independent experiments are shown.

**Figure 6 cells-13-01915-f006:**
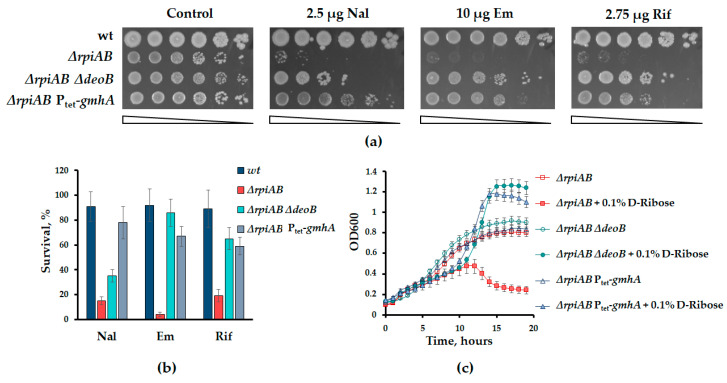
(**а**) Inactivation of the nucleoside catabolism gene (∆*deoB*) and increased biosynthesis of ADP-heptose (P_tet_-*gmhA*) suppress the sensitivity of the *rpiAB* mutant to antibiotics of various mechanisms of action. Representative efficiencies of colony formation of WT (MG1655) and mutant *E. coli* cells in the presence of antibiotics. (**b**) Cell survival was determined by counting cfu and is shown as the mean ± SD from three independent experiments. (**c**) Representative curves demonstrating the suppressive effect of *ΔdeoB* and P_tet_-*gmhA* mutations on sensitivity *ΔrpiAB* to exogenous D-ribose.

**Table 1 cells-13-01915-t001:** Bacterial strains.

Strain	Genotype	Reference
MG1655	F^–^ wild type	Laboratory collection
AM4001	As MG1655 plus *rpiAB*	This work
AM4002	As MG1655 plus *∆rpe*	“
AM4003	As MG1655 plus *∆zwf*	“
AM4005	As AM4001 plus *∆deoB*	“
AM4006	As AM4001 plus P_tet_-*gmhA*	“
AM4007	As MG1655 plus pSoxS’::*lux*	“
AM4008	As AM4001 plus pSoxS’::*lux*	“
AM4009	As AM4002 plus pSoxS’::*lux*	“
AM4010	As AM4003 plus pSoxS’::*lux*	“
**Plasmid**		
pCP20	*FLP*^+^, λ *c*I857^+^, λ *p*_R_ Rep^ts^, Ap^R^, Cm^R^	[18]
pSoxS’::*lux*	pSoxS’:: *luxCDABE,* Ap^R^	[19]

## Data Availability

The original contributions presented in the study are included in the article, further inquiries can be directed to the corresponding authors.

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
