# Peer review of "Disruptions of rpiAB Genes Encoding Ribose-5-Phosphate Isomerases in E. coli Increases Sensitivity of Bacteria to Antibiotics"

_cells, 2024, doi:10.3390/cells13221915_

Round 1

Reviewer 1 Report

Comments and Suggestions for Authors

Review for Journal Cells

Title: Disruption of rpiAB genes encoding ribose-5-phosphate iso-merases in E. coli increases sensitivity of bacteria to antibiotic

Title and abstract

The title matches the aim of the manuscript. The aim of the study is to investigate pentose interconversion of ribose- 9 5-phosphate isomerases RpiA and RpiB and ribulose-5-phosphate epimerase Rpe. The study investigate the effect of deletions of the rpiAB and rpe genes, encoding isomerases and epimerase in E. coli cells on the sensitivity of bacteria to antibiotics and on changes in redox balance.

The findings support deletion of rpiAB leads to a significant increase in the activity of transketalase (Tkt), the key enzyme of the nonoxidative branch of the PPP and increased sensitivity to ribose added in the growth medium. The findings indicate that the induction of unidirectional synthesis of R5P is the cause of super-sensitivity to antibiotics in rpiAB mutant.

Keywords are clear and align with the manuscript. Escherichia coli can be added as the fifth keyword because there can be multiple bacteria for the investigation and authors have chosen E. coli.

Introduction

The writing is clear. The content is laid out clearly in the narratives. The author placed large amount of explanation in the subtitle of Figure 1. Figure 1(a) relates to the narrative on lines 44 to 54 clearly. It is understood that Figure 1(b) is relevant to 1(a). However, the narrative for Figure 1(b) on page 2 is not described until page 5. It is recommended for authors to consider moving both the narrative and figure in proximity for the purpose of clarity.

Material and methods

The writing in material subsection can be presented more succinctly. Authors can consider moving Table 1 to appendix. Instead, it is important for authors to explain why they use the strains and the sourcing of the strains in the narrative.

Authors please provide a sentence to explain the choice of Tukey correlation as opposed to other analysis such as Bonferroni correlation.

Please state the year of GraphPad Prism 9.1.2 software.

Results

Overview: Data compilation

For all figures (For example in Table 4 and 5), if the authors decide to provide diagrammatic representation, please state the mean value such as “Mean values ± SD from at least three independent experiments are shown” collated in a table in the Appendix.

Overview: Statistical analysis

It is a bit hard to appreciate the statistical analysis in the manuscript. As indicated in the method and material, the authors indicated there are statistical analysis of the findings. Please ensure the findings are clearly stated in the writing.

Results

The results in subsection 3.1 and 3.2 are presented clearly most of the time. However, some sentences benefits from rewrite or edit:

“We have found that simultaneous inactivation of the rpiA and rpiB isomerase genes leads to a sharp increase in the sensitivity of E. coli cells to various antibiotics, while cells with a deletion of the rpe gene differ slightly from wild-type cells”(Line 179 to 181). Can authors please rewrite and consider quantify “sharp increase” or “differ slightly” by CFU? Can the output be quantified with colony counts or CFU, and if so, the count should be quantified and compared with statistical analysis.

Figure 2 subtitle and results: Can authors write to make good with the findings? Would authors please consider move the writing in Figure 2’s subtitle to the main body of findings. For instance, colony growth in antibiotics quinolone indicate [Findings]. Colony growth in substrate with [concentration ] ug/mL macrolides indicate [Findings] at 30 minutes and so on.  

For information written in the subtitle in Figure 4, please also re-write in the main body of findings. Figure subtitle can be clearer if used to support the findings. Also please include explanation keys in Figures. For instance, the representation of “PQ” in Figure 4 is likely paraquat and it should be stated as a key in the Figure.

Figure 6. The subtitle included writing from material and method. This should not be included in subtitle. Please delete: “Overnight cultures of indicated E. coli strains were diluted with fresh LB 1:100 and grown to OD600 ≈ 0.5. Antibiotics was added for 30 minuts at concentration 25 µg/ml Nal, 150 µg/ml Em and 25 µg/ml Rif. (c) Representative curves demonstrating the suppressive effect of ΔdeoB and Ptet-gmhA mutations on sensitivity ΔrpiAB to exogenous D-ribose.”

Discussion and findings

There are different styles of writing. Some authors prefer to write in first person writing, while others do not. For this manuscript, it will read better if authors can remove “we found”, “we do not observe” etc. as the manuscript are findings and observations by the authors. So it is redundant to write “we found” etc. Throughout the manuscript, please delete the clauses. Examples are lines 201, 259, 293 for “we found” and line 200 for “we identify”.

For possible rewrite, please consider: “A redox imbalance and restriction of anabolic capacity in the rpiAB mutant is supported by findings of reduced NADPH equivalents, glutathione and ATP pool” to replace “The decrease in the level of NADPH equivalents, reduced glutathione, and the drop in the ATP pool that we found indicate a redox imbalance and restriction of the anabolic capacity  in the rpiAB mutant”.

Now the manuscript aparagraph starts with indent except the last paragraph. Please indent for consistency: “In summary, our study demonstrate that the functional state of ribose-5-phosphate isomerase determines the direction of ribose synthesis in E. coli cells and the exceptional role of this process in potentiating the action of various antibacterial agents. The mechanism of this sensitivity, as well as the possible direct toxic effect of excess R5P on cells, requires further research.” Alternatively, if this is a conclusion, please state so.

Author Response

Comments and Suggestions for Authors

Review for Journal Cells

Title: Disruption of rpiAB genes encoding ribose-5-phosphate iso-merases in E. coli increases sensitivity of bacteria to antibiotic

Title and abstract

The title matches the aim of the manuscript. The aim of the study is to investigate pentose interconversion of ribose- 9 5-phosphate isomerases RpiA and RpiB and ribulose-5-phosphate epimerase Rpe. The study investigate the effect of deletions of the rpiAB and rpe genes, encoding isomerases and epimerase in E. coli cells on the sensitivity of bacteria to antibiotics and on changes in redox balance.

The findings support deletion of rpiAB leads to a significant increase in the activity of transketalase (Tkt), the key enzyme of the nonoxidative branch of the PPP and increased sensitivity to ribose added in the growth medium. The findings indicate that the induction of unidirectional synthesis of R5P is the cause of super-sensitivity to antibiotics in rpiAB mutant.

Keywords are clear and align with the manuscript. Escherichia coli can be added as the fifth keyword because there can be multiple bacteria for the investigation and authors have chosen E. coli.

Escherichia coli added to the list of keywords

Introduction

The writing is clear. The content is laid out clearly in the narratives. The author placed large amount of explanation in the subtitle of Figure 1. Figure 1(a) relates to the narrative on lines 44 to 54 clearly. It is understood that Figure 1(b) is relevant to 1(a). However, the narrative for Figure 1(b) on page 2 is not described until page 5. It is recommended for authors to consider moving both the narrative and figure in proximity for the purpose of clarity.

Added link to figure 1b in line 49

Material and methods

The writing in material subsection can be presented more succinctly. Authors can consider moving Table 1 to appendix. Instead, it is important for authors to explain why they use the strains and the sourcing of the strains in the narrative.

Authors please provide a sentence to explain the choice of Tukey correlation as opposed to other analysis such as Bonferroni correlation.

We used Tukey's test as the most popular for multiple comparisons procedure.

Please state the year of GraphPad Prism 9.1.2 software.

Corrected

Results

Overview: Data compilation

For all figures (For example in Table 4 and 5), if the authors decide to provide diagrammatic representation, please state the mean value such as “Mean values ± SD from at least three independent experiments are shown” collated in a table in the Appendix.

Overview: Statistical analysis

It is a bit hard to appreciate the statistical analysis in the manuscript. As indicated in the method and material, the authors indicated there are statistical analysis of the findings. Please ensure the findings are clearly stated in the writing.

The results of the statistical analysis are presented in the Appendix A in the table 2, 3 and 4

Results

The results in subsection 3.1 and 3.2 are presented clearly most of the time. However, some sentences benefits from rewrite or edit:

“We have found that simultaneous inactivation of the rpiA and rpiB isomerase genes leads to a sharp increase in the sensitivity of E. coli cells to various antibiotics, while cells with a deletion of the rpe gene differ slightly from wild-type cells”(Line 179 to 181). Can authors please rewrite and consider quantify “sharp increase” or “differ slightly” by CFU? Can the output be quantified with colony counts or CFU, and if so, the count should be quantified and compared with statistical analysis.

Figure 2b is a quantitative assessment of cell survival. The figure is supplemented with statistical data according to your comments.

Figure 2 subtitle and results: Can authors write to make good with the findings? Would authors please consider move the writing in Figure 2’s subtitle to the main body of findings. For instance, colony growth in antibiotics quinolone indicate [Findings]. Colony growth in substrate with [concentration ] ug/mL macrolides indicate [Findings] at 30 minutes and so on.  

The corresponding text was corrected

For information written in the subtitle in Figure 4, please also re-write in the main body of findings. Figure subtitle can be clearer if used to support the findings. Also please include explanation keys in Figures. For instance, the representation of “PQ” in Figure 4 is likely paraquat and it should be stated as a key in the Figure.

The corresponding text was corrected

Figure 6. The subtitle included writing from material and method. This should not be included in subtitle. Please delete: “Overnight cultures of indicated E. coli strains were diluted with fresh LB 1:100 and grown to OD600 ≈ 0.5. Antibiotics was added for 30 minuts at concentration 25 µg/ml Nal, 150 µg/ml Em and 25 µg/ml Rif. (c) Representative curves demonstrating the suppressive effect of ΔdeoB and Ptet-gmhA mutations on sensitivity ΔrpiAB to exogenous D-ribose.”

The corresponding text was corrected

Discussion and findings

There are different styles of writing. Some authors prefer to write in first person writing, while others do not. For this manuscript, it will read better if authors can remove “we found”, “we do not observe” etc. as the manuscript are findings and observations by the authors. So it is redundant to write “we found” etc. Throughout the manuscript, please delete the clauses. Examples are lines 201, 259, 293 for “we found” and line 200 for “we identify”.

The corresponding text was corrected

For possible rewrite, please consider: “A redox imbalance and restriction of anabolic capacity in the rpiAB mutant is supported by findings of reduced NADPH equivalents, glutathione and ATP pool” to replace “The decrease in the level of NADPH equivalents, reduced glutathione, and the drop in the ATP pool that we found indicate a redox imbalance and restriction of the anabolic capacity in the rpiAB mutant”.

The corresponding text was corrected

Now the manuscript aparagraph starts with indent except the last paragraph. Please indent for consistency: “In summary, our study demonstrate that the functional state of ribose-5-phosphate isomerase determines the direction of ribose synthesis in E. coli cells and the exceptional role of this process in potentiating the action of various antibacterial agents. The mechanism of this sensitivity, as well as the possible direct toxic effect of excess R5P on cells, requires further research.” Alternatively, if this is a conclusion, please state so.

Corrected

Reviewer 2 Report

Comments and Suggestions for Authors

Introduction

Written well.

At the end of the introduction section, mention why this study is important.

Methods

In table 1 include plasmids used in this study.

Explain the mutant construction protocol briefly for better understanding of readers.

Section 2.2.- Mention the concentration of antibiotics.

Figures:

Figure 4a, Figure 5a, c, and d, and Figure 6c include error bars in the growth curves.

In figure 5, revise annotations (a, b, c, and d).

Discussion

The Discussion section currently lacks citations, and I recommend that the authors enhance the Discussion section by citing appropriate references to support their findings and provide a comprehensive context for their research.  

Include a separate conclusion section.

Author Response

Comments and Suggestions for Authors

Introduction

Written well.

At the end of the introduction section, mention why this study is important.

The corresponding text was corrected

Methods

In table 1 include plasmids used in this study.

Corrected

Explain the mutant construction protocol briefly for better understanding of readers.

Section 2.2.- Mention the concentration of antibiotics.

Corrected

Figures:

Figure 4a, Figure 5a, c, and d, and Figure 6c include error bars in the growth curves.

In figure 5, revise annotations (a, b, c, and d).

Corrected

Discussion

The Discussion section currently lacks citations, and I recommend that the authors enhance the Discussion section by citing appropriate references to support their findings and provide a comprehensive context for their research.

  Corrected

Include a separate conclusion section.

The separate conclusion section included
